# M⁴olGen: Multi-Agent, Multi-Stage Molecular Generation under Precise Multi-Property Constraints

## Abstract

Generating molecules that satisfy precise numeric constraints over multiple physicochemical properties is critical and challenging. Although large language models (LLMs) are expressive, they struggle with precise multi-objective control and numeric reasoning without external structure and feedback. We introduce **M⁴olGen**, a fragment-level, retrieval-augmented, two-stage framework for molecule generation under multi-property constraints. **Stage I: Prototype generation**: a multi-agent reasoner performs retrieval-anchored, fragment-level edits to produce a candidate near the feasible region. **Stage II: RL-based fine-grained optimization**: a fragment-level optimizer trained with Group Relative Policy Optimization (GRPO) applies one- or multi-hop refinements to explicitly minimize the property errors toward our target while regulating edit complexity and deviation from the prototype. A large, automatically curated dataset with reasoning chain of fragment edits and measured property deltas underpins both stages, enabling deterministic, reproducible supervision and controllable multi-hop reasoning. Unlike prior work, our framework better reasons about molecules by leveraging fragments and supports controllable refinement toward numeric targets. Experiments on generation under three property constraints (QED, LogP, and molecular weight) show consistent gains in validity and precise satisfaction of multi-property targets, outperforming strong LLMs and graph-based algorithms.

## 1 Introduction

Generating molecules that satisfy precise numeric constraints is a fundamental and critical task in scientific discovery, with applications in drug development, materials design, de novo design and molecular property optimization (Sanchez-Lengeling & Aspuru-Guzik, 2018; Fromer & Coley, 2023). Optimizing compounds to meet numeric multi-property targets improves real development outcomes with desired attributes (Wager et al., 2016). Much of the molecular generation literature treats molecular discovery as maximizing one or a few surrogate properties, rather than matching user-specified numerical targets; approaches that offer precise, simultaneous control over multiple properties remain scarce. Recent generative models condition on desired magnetic density, bandgap, and bulk modulus along with chemistry and symmetry, demonstrating the feasibility of property-conditioned generation (Zeni et al., 2025; Ding et al., 2024). We focus on small-molecule discovery, where practical constraints are drug-centric and include drug-likeness (QED), lipophilicity (logP), and molecular weight (MW)—properties that shape permeability, exposure, and overall developability (Bickerton et al., 2012; Giaginis et al., 2018). While these are simplified surrogates, they are (i) fast and reproducible to evaluate (enabling large-scale training and ablations), (ii) continuous and numeric, which is essential for testing precise multi-objective control, and (iii) standardized across open benchmarks, supporting fair comparison. Our goal in this paper is to validate a multi-agent, numerically conditioned generation framework under verifiable, compute-efficient proxies; in principle the same machinery can swap in richer oracles as we scale to more realistic discovery settings. We aim to introduce a new solution handling molecular generation with specific property requirements by enabling precise, multi-property control at specified numeric targets.

Large language models (LLMs) have shown promise and have become more and more popular in molecular generation (Ramos et al., 2025; Wang et al., 2025), but struggle to reason over multiple

numeric targets simultaneously (Li et al., 2025). This difficulty stems from LLMs' limited numerical target reasoning and insufficient domain-grounded reasoning. To bridge this gap, reinforcement learning (RL) is increasingly used alongside to inject explicit, objective-driven feedback that guides editing actions during molecular generation. However, RL methods such as REINVENT (Loeffler et al., 2024), while capable of handling multi-property objectives, typically require fine-tuning for each target vector, making them time- and compute-intensive at scale.

To address these challenges, we introduce $M^4$olGen, a **M**ulti-stage, **M**ulti-agent framework for **M**ulti-property-constrained **Mol**ecular **Gen**eration. Our core idea is a unified formulation that casts numeric targets as a verifiable error-to-target objective over an actionable fragment-edit space, so progress is measurable at every step and complexity is controllable. Our framework consists of two stages. Stage I performs retrieval-augmented, fragment-level prototyping: a local reasoning agent iteratively edits fragments, guided by in-distribution exemplars and numeric feedback from chemistry tools (RDKit (Landrum)), to place the candidate near the feasible region. Here fragments are defined as building blocks by breaking molecules along synthetically accessible bonds through RDkit. Stage II delivers fine-grained, multi-hop refinement with a fragment-level optimizer trained via Group Relative Policy Optimization (GRPO) (Shao et al., 2024); it explicitly minimizes the error-to-target across properties, and crucially lets us control structural complexity and deviation from the original candidate, not merely meeting requirements. By grounding updates in verifiable property oracles and reward signals, this optimizer overcomes the limitations of LLMs operating solely on domain knowledge stored in LLM weights, enabling reliable numerical control.

To fine-tune the optimizer, we construct a large dataset of more than 2 million molecules decomposed into BRICS (Degen et al., 2008) fragments along with their corresponding properties. From this dataset, we derive a neighbor relational dataset of 1.17 million pairs for controllable reasoning automatically. Each molecule in this dataset is paired with an explicit one-hop neighbor list: molecules that differ by exactly one fragment (add, remove, or replace) and that pass the RDKit validity and edit sanity checks. By chaining these one-hop moves, we gradually grow neighbor forests from any starting molecule. These structures enable long, controllable reasoning chains: we can choose the depth and branching to regulate structural complexity and deviation from the original, build curricula that move from coarse adjustments to fine tuning, sample forward and reverse paths to supervise multi-hop optimization, error-to-target feedback at every step. We demonstrate that this architecture markedly improves adherence to numeric multi-property constraints and surpasses prior LLM-based methods by large margins.

In summary, we contribute (i) $M^4$olGen, a molecular generation framework that couples retrieval-augmented prototyping with GRPO-based fragment-level optimization to achieve *exact numeric control* over multiple properties; (ii) a scalable *multi-hop* refinement mechanism that boosts output quality while explicitly regulating edit complexity and deviation from the starting structure; (iii) a public dataset of $\sim$2.95M molecules with BRICS fragment annotations and a neighbor set of $\sim$1.17M single-edit pairs that enable fragment-level learning and controllable reasoning; and (iv) comprehensive experiments and ablations demonstrating state-of-the-art normalized total error with clear additive gains from each component.

## 2 RELATED WORK

**Molecular Generation with Property Control.** Deep generative models have been widely applied to molecular design, leveraging graph or sequence-based representations such as SMILES. Early works include VAEs (Gómez-Bombarelli et al., 2016) and GANs such as MolGAN (Cao & Kipf, 2018), followed by graph-based models like GCPN (You et al., 2018), GraphAF (Shi et al., 2020), and MoFlow (Zang & Wang, 2020).STGG+ (Jolicoeur-Martineau et al., 2025), which is extended from Spanning Treebased Graph Generation, shows promising performance in multi-objective optimization. Reinforcement learning approaches (e.g., MolDQN (Zhou et al., 2018)) enable property-driven optimization, often with multi-objective extensions for QED, LogP, and SA. However, these single-agent methods struggle to exactly satisfy multiple numeric constraints, reflecting exploration–exploitation trade-offs.

**LLMs for Molecular Design and Reasoning.** Large language models (LLMs) such as ChemGPT (Frey et al., 2023), ChemBERTa (Chithrananda et al., 2020), MolT5 (Edwards et al., 2022), and

Chemformer (Irwin et al., 2021) capture chemical syntax and semantics, enabling general-purpose molecular generation. While expressive, they remain limited in precise numerical reasoning and property control. Chain-of-Thought prompting (Wei et al., 2022) improves interpretability and multi-step reasoning in LLMs, and analogous strategies have been suggested for molecules (Jin et al., 2024; Jang et al., 2024; Zheng et al., 2024), aligning with human-in-the-loop frameworks. Yet, exact satisfaction of multiple physicochemical constraints remains challenging. Recent work such as Instruction Multi-Constraint Molecular Generation (Zhou et al., 2025) demonstrates that LLMs can satisfy multiple property constraints through teacher–student supervised training and interval-based conditioning. However, these methods primarily operate within bounded property ranges and are not based on reinforcement learning for multi-objective optimization.

**Multi-Agent Planning and Reasoning in Molecule Design.** Agent-based systems have long been studied in robotics, distributed AI, and resource allocation (Wooldridge, 2009; Weiss, 1999). In molecule design, however, most AI-driven approaches remain single-agent, where a single generative model is guided by property predictors. Recent work has begun to explore multi-agent systems that decompose the design process into specialized roles, such as generation, property evaluation, and refinement, by enabling cooperation or hierarchical coordination, these systems can improve exploration efficiency and controllability. For example, recent works like Prompt-to-Pill (Vichentijevikj et al., 2025), ROBIN (Ghareeb et al., 2025), DrugAgent (Liu et al., 2024), Honeycomb(Zhang et al., 2024) and ChemCrow (M. Bran et al., 2024) have demonstrated the power of this multi-agent paradigm. Building on this line of research, we introduce a retrieval-augmented multi-agent reasoner that iteratively constructs locally optimal prototypes before refinement. This allows our system to combine in-distribution retrieval with domain knowledge to improve controllability under numeric property constraints.

**Policy Optimization for Multi-Property Objectives.** Reinforcement learning provides a foundation for molecular optimization. Classical policy-gradient methods such as REINFORCE (Williams, 2004) and proximal policy optimization (PPO) (Schulman et al., 2017) have been adapted to molecule design. MolDQN (Zhou et al., 2018), for example, leverages Q-learning for multi-objective optimization. However, these approaches face difficulties in balancing multiple numeric objectives precisely. Group Relative Policy Optimization (GRPO) (Shao et al., 2024; Zhang et al., 2025), originally introduced for preference-based learning and RLHF, optimizes policies via group-relative advantages that reward candidates outperforming their peers. While GRPO and its modified versions are well known for strengthening LLM reasoning (DeepSeek-AI et al., 2025), we are the first to adapt it to numerically conditioned generation, integrating fragment-level refinement and controllable multi-hop optimization within the generation loop. This yields a principled reinforcement-learning framework for satisfying numeric multi-property targets.

## 3 METHODOLOGY

We propose $M^4$olGen, shown in Figure 1, a multi-stage, goal-conditioned framework for constrained molecular generation that casts numeric targets (QED, LogP, MW) as a verifiable distance-to-target objective over an actionable fragment-edit space. Stage I performs retrieval-augmented prototyping: a local reasoner edits fragments using in-distribution exemplars and RDKit feedback to place a candidate near the feasible region. Stage II applies a GRPO-trained fragment-level optimizer in a multi-hop manner to minimize the distance-to-target while regulating edit complexity and deviation from the starting structure. Trained on a large, property-annotated neighbor dataset, $M^4$olGen generalizes across target tuples and delivers precise, simultaneous control of QED, LogP, and MW (Molecular Weight) and shows capabilities that prompt-only LLMs struggle to achieve due to limited numerical reasoning.

### 3.1 STAGE I: PROTOTYPE GENERATION WITH RETRIEVE-AUGMENTED MULTI-AGENT REASONING

The objective of Stage I is to generate a chemically valid prototype $m_{local}$ that serves as a high-quality starting point for numeric optimization. This is accomplished via a collaborative multi-agent framework that decomposes the input query, retrieves similar molecules from a large database, and incrementally proposes fragment-level edits based on domain knowledge.

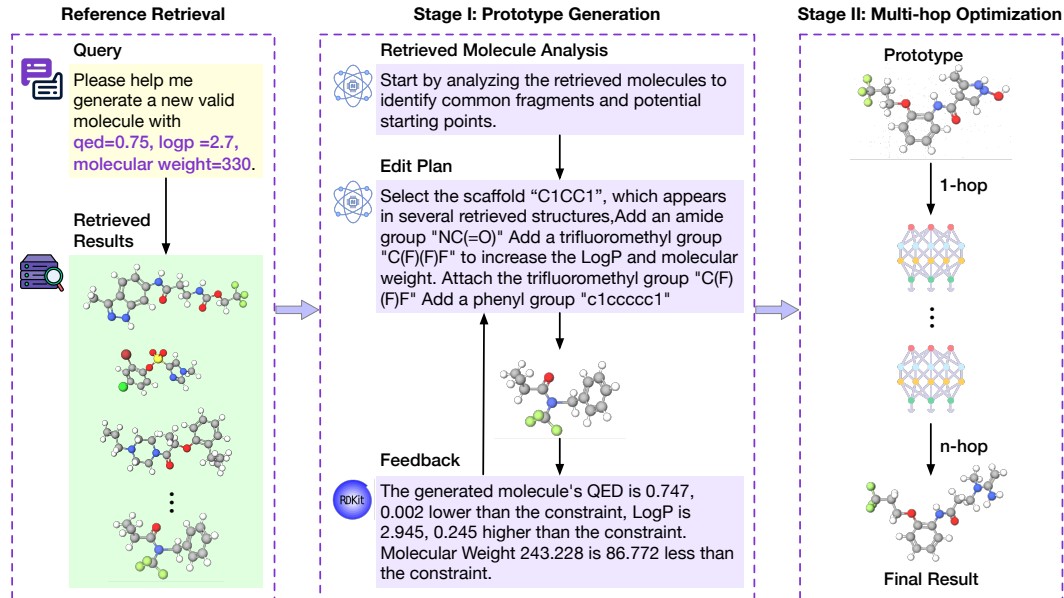

Figure 1: The flow chart of M$^4$olGen. The first two blocks involve **Retrieval and Prototyping**, where molecular candidates are first retrieved based on the given constraints (QED, LogP, MW) and then analyzed by a local reasoner to extract constraints, analyze retrieved molecules, and propose an editing plan based on evaluator's feedback to generate prototypes iteratively. The third block describes **Multi-Hop Optimization**, where the prototypes are optimized through one-hop and n-hop controllable editing steps by the molecule optimizer trained by GRPO.

**Query interpretation.** Given a natural-language request $q$ (e.g., *"Generate a molecule with QED=0.72, LogP=−1.8, MW=310"*), this module extracts the exact numeric targets for each property and returns a target property vector

$$\mathbf{p}_{\text{tgt}} = \big(p_{\text{QED}}, p_{\text{LogP}}, p_{\text{MW}}\big), \quad p_{\text{QED}} \in [0,1], \ p_{\text{LogP}} \in \mathbb{R}, \ p_{\text{MW}} > 0. \tag{1}$$

We use $\mathbf{p}$ *for "properties"* and the subscript "tgt" to denote targets. A rule-based parser identifies numeric constraints and synonyms (e.g., "molecular weight", "MW").

**Reference retrieval.** Given the target property vector $\mathbf{p}_{\text{tgt}}$, we query a large molecule corpus $\Omega$ to obtain a set of *reference molecules* that lie close to the targets under per–property tolerances:

$$\mathcal{M} = \big\{ m \in \Omega \ : \ |p_i(m) - p_{i,\text{tgt}}| \leq \epsilon_i \ \forall i \in \{\text{QED}, \text{LogP}, \text{MW}\}\big\}. \tag{2}$$

Here $p_i(m)$ denotes the $i$-th property of molecule $m$ (computed via RDKit), and $\epsilon_i$ are small, property-specific tolerant ranges (e.g., $\pm0.05$ for QED (0–1 scale), $\pm0.5$ for $LogP$ (small medicinally meaningful shift), and $\pm25$ Da for MW.). They are chosen to be tight enough to keep the references in-distribution yet broad enough to ensure sufficient references. The retrieved references are then used to *anchor* Stage I: they provide in-distribution exemplars that guide fragment-level edits, constrain the search toward the feasible region, and seed candidate/neighbor structures consumed by the multi-hop optimizer in Stage II.

**Prototype reasoner.** This LLM-driven module proposes stepwise, fragment-level edits to turn an initial seed (either "start from scratch" or molecules sampled from the reference set $\mathcal{M}$) into a high-quality *prototype* close to the target. At iteration $t$, the reasoner selects an action $a_t \in \{\texttt{replace}, \texttt{add}, \texttt{remove}\}$ and applies it to obtain a new intermediate molecule along with previous trajectory

$$m_t = \text{Edit}(m_{t-1}; a_t), \qquad m_t \in \mathcal{M}_{\text{valid}}, \tag{3}$$

where $\mathcal{M}_{\text{valid}}$ denotes RDKit-parseable structures that pass basic valence and sanity checks. Decisions are guided by three information sources: (i) *reference molecules* $\mathcal{M}$ retrieved near the target, (ii) an *experience pool* of prior edits (neighbor pairs/trees) summarizing successful local transformations, and (iii) *property feedback* (QED/LogP/MW) computed by RDKit on every candidate.

The reasoner stops early when the distance-to-target falls below a threshold $\tau$ or when a maximum number of steps $T_{\max}$ is reached.

**Validity and error estimation.** Given the current prototype $m_{\text{local}}$, we compute per–property deviations from the targets

$$\Delta_{\text{i}}(m) = \big| p_i(m_{\text{local}}) - p_{i,\text{tgt}} \big|, \ i \in \{\text{QED, LogP, MW}\} \tag{4}$$

and aggregate them into a distance-to-target objective $E(m) = \sum_i w_i \, \Delta_i(m)$ with property-specific weights. These errors are fed into the Stage II optimizer prompt to enable targeted refinement.

**Stage I objective.** Formally, Stage I seeks a valid prototype along the reasoning trajectory $\mathcal{G} = \{m_0, \ldots, m_T\}$ that minimizes the distance-to-target:

$$m_{\text{local}} = \arg \min_{m \in \mathcal{G} \cap \mathcal{M}_{\text{valid}}} \sum_{i \in \{\text{QED,LogP,MW}\}} w_i \big| p_i(m) - p_{i,\text{tgt}} \big|. \tag{5}$$

The algorithm is stated in Appendix A.1. This stage reliably moves the candidate into the feasible region by leveraging relevant molecules, past experience, and tool feedback. However, a multi-agent reasoner that is not further trained has a performance limitation on fine-grained, precise multi-property control. Stage II addresses this by applying a GRPO-trained, fragment-level optimizer in a controlled multi-hop fashion to further reduce the total error $E(m)$ while regulating edit complexity and deviation from the starting structure.

## 3.2 STAGE II: FRAGMENT-LEVEL OPTIMIZATION VIA GRPO (MULTI-HOP EXTENSION)

While Stage I reliably moves a candidate near the feasible region, precise control of multiple numeric properties (e.g., QED, LogP, MW) remains difficult for text-only planning because LLMs have difficulty dealing with numeric-related design and lack a mechanism to explicitly minimize the distance to target values. Our insight is to treat refinement as an optimization problem over an actionable fragment-edit space with fast, verifiable feedback from chemistry oracles. We therefore train an optimization policy with **GRPO** (Group Relative Policy Optimization) (DeepSeek-AI et al., 2025) because its group-wise, rank-based updates are stable and sample-efficient without ground-truth demonstrations, and because it can directly optimize a reward that faithfully encodes the numeric targets. RDKit oracles provide the property feedback at each step, making the reward precise and inexpensive to evaluate.

**Fragmentization and action space.** Let $m_0 := m_{\text{local}}$ be the prototype from Stage I. We decompose molecules into chemically meaningful building blocks using **BRICS** (Break Retrosynthetically Interesting Chemical Substructures) (Degen et al., 2008), a rule-based scheme that cuts retrosynthetically plausible bonds formed or broken during synthetic processes, leading to fragments that are synthetically accessible and chemically meaningful. This yields fragments that support localized edits, preserve validity, and keep the search space tractable where $\Phi(m)$ is the fragment set for molecule $m$ and $f$ are the fragments:

$$\Phi(m) = \{ f_1, \ldots, f_k \}. \tag{6}$$

At hop $h \in \{1, \ldots, H\}$, the optimizer selects one fragment-level action $a_h \in \{\texttt{add}, \texttt{remove}, \texttt{replace}\}$ and applies it to obtain a new candidate

$$m_h = \text{Edit}(m_{h-1}; a_h), \qquad m_h \in \mathcal{M}_{\text{valid}}, \tag{7}$$

where $\mathcal{M}_{\text{valid}}$ denotes RDKit-parseable structures that pass basic valence and sanity checks. A hop budget $H$ controls structural complexity and deviation from the starting structure.

**Optimizer and input representation.** Our optimizer $\mathcal{O}_\phi$ is a sequence model (an LLM policy) fine-tuned with GRPO on our neighbor-pair corpus of single-fragment edits. Following Guevorguian et al. (2024), we extend the tokenizer with `<SMILES>`, `</SMILES>`, `<QED>`, `</QED>`, `<LogP>`, `</LogP>`, `<MW>`, `</MW>` so that molecules and targets are explicit in the prompt. At each hop, the policy conditions on $(m_{h-1}, \Phi(m_{h-1}), \mathbf{p}_{\text{tgt}})$ and proposes one edited molecule; after $H$ hops we return $m^* := m_H$.

**Reward and GRPO objective.** We define a distance-to-target objective and convert it to a scalar reward using fast RDKit oracles:

$$E(m) = \sum_{i \in \{\text{QED,LogP,MW}\}} w_i \,\big| p_i(m) - p_{i,\text{tgt}} \big|, \qquad R_{\text{prop}}(m) = 1 - E(m). \tag{8}$$

The full reward combines format, property, diversity, and validity terms:

$$R(m) = \underbrace{r_{\text{format}}(m)}_{\text{valid SMILES / instruction}} + \underbrace{R_{\text{prop}}(m)}_{\text{scaled property match}} - \underbrace{r_{\text{repeat}}(m)}_{\text{repetition penalty}} - \underbrace{r_{\text{invalid}}(m)}_{\text{RDKit parse / valence penalty}}. \tag{9}$$

Here $w_i$ are *weights* that balance units and priorities; $\mathrm{EditCost}$ optionally regularizes complexity (e.g., hop count or similarity). GRPO samples a group of candidates, ranks them by $R(m)$, converts ranks to normalized rewards, and updates the policy to increase the likelihood of higher-ranked edits while discouraging weaker ones. This group-relative signal is robust under noisy rewards and directly steers the policy toward exact numeric targets without lossy surrogate models.

**Multi-hop refinement and control.** Applying the optimizer in a controlled multi-hop manner enables gradual, interpretable refinement: small, local edits accumulate to tighten requirement satisfaction, while the hop budget and regularizers bound complexity and deviation from the prototype. In practice, a modest $H$ suffices to reliably reduce $E(m)$ thanks to fragment locality and fast RD-Kit evaluation, and the same mechanism supports adaptive planning and curriculum-style difficulty scaling during training and evaluation.

### 3.3 Automated Synthesis of Reasoning Dataset

To train an optimizer that not only generates strings, but reasons about edits, we require a corpus that (i) couples each molecule with reliable physicochemical properties, (ii) exposes an actionable fragment space (fragments and how they connect), and (iii) provides neighbor relations so we can supervise single-step edits and assemble multi-hop reasoning chains. This enables reward-driven refinement under exact numeric targets.

We merge all the molecules from ZINC (Irwin & Shoichet, 2005), CHEMBL (Gaulton et al., 2012) and MOSES (Polykovskiy et al., 2020) together, filter and delete the duplicates. From each molecule we obtain its SMILES, molecular formula, QED, logP, logS, and molecular weight computed with RDKit. We further derive a fragment decomposition and an inter-fragment connectivity map (identifying the bonds between fragments). The final dataset contains 2,945,596 molecules and, to the best of our knowledge, is the largest resource coupling molecular properties with fragment-based structural annotations.

Starting from our unified corpus, we build a reasoning-ready resource through an automated pipeline: (i) **standardize & deduplicate** molecules via RDKit canonical SMILES, neutralize, and enforce valence/aromaticity sanity checks; (ii) **annotate properties** (QED, LogP, LogS, MW) with RDKit; (iii) **fragmentize** each molecule with BRICS to obtain a fragment multiset $\Phi(m)$ and an inter-fragment connectivity map (which fragments are joined and at which bonds), yielding an actionable edit space; (iv) **construct neighbor pairs** by scanning for molecules that differ by exactly one fragment-level edit (`add`/`remove`/`replace`), while enforcing edit sanity (e.g., element-count conservation for `replace`) and RDKit validity for the edited product; and (v) **label supervision** by recording the edit type, edited fragments, and signed property deltas ($\Delta\text{QED}, \Delta\text{LogP}, \Delta\text{MW}$), plus the distance-to-target objective used by our optimizer. This process yields a **neighbor-pair corpus of ∼1,171,193 single-edit pairs**. For each molecule we also materialize its **1-hop neighbor list** based on fragment multiset edit distance, from which we grow neighbor trees/forests. These structures serve two roles: they seed *retrieval-anchored prototyping* in Stage I and provide *experience-based, reward-compatible supervision* for GRPO in Stage II, enabling controllable multi-hop refinement under exact numeric targets. Each entry is formatted as a natural language prompt with a one-step edit answer, e.g.:

> Given the intermediate molecule SMILES `<SMILES>O=C(NCc1nccc2ccccc12)c1c cc[nH]c1=O</SMILES>`, which is composed of fragments ['C()=O', 'N', 'C', 'c1nccc2ccccc12', 'c1ccc[nH]c1=O']. Propose a single replace, add or remove step on fragment level that makes the new molecule's QED `<QED>0.146</QED>` lower, LogP `<LogP>0.366</LogP>` higher, and Molecular Weight `<MW>53.068</MW>` lower.

Replace `c1ccc[nH]c1=O` with `c1nc2nc(C)cc(C)n2n1` to form `<SMILES>Cc1cc(C)n2nc(C(=O)NCc3nccc4ccccc34)nc2n1</SMILES>`.

GRPO itself does not need any ground truth for editing, but all property changes are still derived from real data to preserve distribution realism.

## 4 EXPERIMENT

Our studies are designed to validate the four core claims from the introduction and to do so with minimal assumptions. **(C1) Precise multi-property control:** we benchmark M$^4$olGen against strong LLMs and graph methods under identical compute budgets, reporting per–property MAE and a normalized total error to demonstrate simultaneous control of QED/LogP/MW. **(C2) Necessity and effectiveness of the two-stage design:** we perform ablations that toggle retrieval in Stage I and vary the GRPO optimizer hops (1/2/3), to show that retrieval-augmented prototyping plus multi-hop refinement is required for tight numeric alignment. **(C3) Generalization without per-target retraining & controllable edit complexity:** we uniformly sample 100 target tuples across admissible ranges, run 10 trials per tuple/baseline (best-of-10 under a fixed budget), and analyze performance as a function of hop budget, establishing broad generalization and explicit control of deviation from the prototype.

### 4.1 EXPERIMENTAL SETUP

**Training Details** In Stage I, we employ GPT-4o(OpenAI, 2024b) as the prototype-reasoning LLM, given its strong instruction-following performance and broad commercial adoption; other capable LLMs can be substituted without changing the framework. For the stage-2 training, we select ChemDFM-v1.5-8B (Zhao et al., 2025) as the base model, which achieves overall great performance among chemical generation tasks. We first train ChemDFM-v1.5-8B for 5000 steps with supervised fine-tuning for cold start. This can accelerate the convergence speed for the following GRPO training since the reward function can get effective feedback sooner than randomly exploring the format first. Then the model is trained for 37,500 steps with GRPO. The scalars we choose for the reward function are $\alpha_q$=1, $\alpha_l$=6, $\alpha_w$=100, as we consider error values 1 in QED, 6 in LogP and 100 in MW as the maximum thresholds. These scalars are flexible to tune depending on personal usage. The invalidity penalty and wrong format penalty are both -10 while the repetition penalty is accumulated by 0.1 for each time. At each step, we sample 8 candidates using stochastic decoding (temperature $T = 1.0$, top-$p = 0.9$, top-$k = 50$). The model was trained to convergence on a single NVIDIA A100 (40 GB).

**Baselines** We aim to investigate the power of LLMs for generating new molecules under precise constraints. Thus, most of the baselines we choose are LLMs. In the LLM-based solutions, we have gpt-4.1 (OpenAI, 2025), Gemini-Flash (Google, 2025), claude-haiku (Anthropic, 2024), gpt-4o-2024-05-13(latest version) (OpenAI, 2024a), SmileyLlama-8B (Cavanagh et al., 2025) and DrugAssist-7B (Ye et al., 2023). They cover most commonly used comerical models and generation-oriented fine-tuned chemical LLMs including SFT(Supervised Fine-Tuning) and DPO(Direct Preference Optimization) technique. In addition to LLM baselines, we also try commonly used graph-based and mixed algorithms. STGG+ (Jolicoeur-Martineau et al., 2025), which is an autoregressive generative model that uses spanning tree-based graph generation to perform multi-property conditional generation and claim to be the state-of-art for multi-objective conditional generation. We also include a graph genetic algorithm (Graph GA) (Jensen, 2019), which requires target-specific optimization; for each target tuple we run it from scratch with oracle calls of 500 and 1000(denoted GA-500 and GA-1000).

**Metrics** We compute the QED, LogP and Molecular Weight and compare them with the target to get the MAE (mean absolute error) for each property. It is commonly used among molecular generation and design benchmarks (Wu et al., 2018). However, for multi-objective optimization task like what we aim to address, it is necessary to have a normalized total error so that we can directly determine which candidate is better. Different properties have different ranges, and individual properties need to be normalized to the same range for multi-objective molecule design (Luukkonen et al., 2023). Therefore, we normalize the error by dividing QED error by 1, LogP error by 10 and MW error by 700 since QED range is from 0 to 1, LogP range is from -10 to 10 and most in-distribution MW range is from 100 to 800. Note that the normalizer for each error can be tuned

when dealing with custom distribution or specific-property-preferred generation. Besides the whole range normalization, we also add the scalars we used for the optimizer's GRPO training(1 for QED, 6 for LogP and 100 for MW). Beyond accuracy, we assess **set quality**. *Uniqueness* is the fraction of distinct molecules among the outputs (measured via canonical SMILES), indicating the absence of duplicates. *Diversity* measures how dissimilar the set is on average, computed from ECFP4 fingerprints (Rogers & Hahn, 2010) with Tanimoto similarity (higher diversity means broader exploration of chemical space).

Table 1: Error metrics across methods (lower is better). Best per column in **bold**; second best underlined.

| Method | QED err | logP err | MW err | Scaled total err | Norm. total err | Diversity | Uniqueness |
|---|---|---|---|---|---|---|---|
| **LLMs** | | | | | | | |
| gpt-4.1 | 0.115 | 0.697 | 49.182 | 0.723 | 0.255 | 0.823 | 1.0 |
| gpt-4o-2024-05-13 | 0.115 | 0.847 | 60.203 | 0.858 | 0.285 | 0.868 | 1.0 |
| Gemini-2.5-Flash | 0.078 | 0.974 | 86.174 | 1.102 | 0.299 | 0.842 | 0.97 |
| Claude-3.7-Sonnet | 0.104 | 1.025 | 39.583 | 0.671 | 0.263 | 0.868 | 1.0 |
| Claude-3.5-haiku | 0.117 | 1.174 | 46.904 | 0.782 | 0.301 | 0.791 | 1.0 |
| SmileyLlama-8B | 0.374 | 2.385 | 196.235 | 2.734 | 0.893 | 0.853 | 1.0 |
| DrugAssist-7B | 0.176 | 2.44 | 165.047 | 2.233 | 0.656 | 0.845 | 0.38 |
| **Graph algorithms** | | | | | | | |
| STGG-50times | **0.050** | 0.566 | 63.917 | 0.784 | 0.198 | 0.876 | 1.0 |
| STGG-10times | 0.100 | 0.754 | 52.760 | 0.753 | 0.306 | 0.879 | 1.0 |
| Graph GA-500 | 0.131 | 0.806 | 15.016 | 0.415 | 0.233 | 0.884 | 1.0 |
| Graph GA-1000 | 0.123 | 0.529 | **7.95** | 0.291 | 0.187 | **0.886** | 1.0 |
| **Our methods** | | | | | | | |
| 1-hop | 0.130 | 0.423 | 10.404 | 0.305 | 0.187 | 0.879 | 1.0 |
| 2-hop | 0.111 | 0.339 | 10.489 | 0.272 | 0.160 | 0.883 | 1.0 |
| 3-hop | 0.103 | **0.284** | 9.799 | **0.249** | **0.146** | 0.884 | 1.0 |

## 4.2 RESULTS AND ANALYSIS

**Protocol.** GRPO is ground-truth–free and reward-based, so performance is not tied to a particular training distribution. To test generalization, we uniformly sample 100 target tuples $(\mathrm{QED}, \log P, \mathrm{MW})$ across admissible ranges. For each tuple and each baseline we run 10 independent trials under the same compute budget and report the *best-of-10*. For STGG+ we consider two sampling budgets ($10\times$ and $50\times$). Across settings, our normalized total error (NTE) decreases monotonically with hop count.

**Main results.** Table 1 compares LLMs, graph baselines, and our method. Our best configuration (3-hop) attains the lowest **NTE** (normalized total error) of **0.146**, improving over the strongest commercial model (GPT-4.1, 0.255) by **42.7%** and over the best non-ours baseline (Graph GA-1000, 0.187) by **21.9%**. Per metric, we obtain the best **logP** error (**0.284**) and the second-best **MW** error (9.799; GA-1000 is **7.95**). Relative to STGG-50×, our 3-hop reduces logP from 0.566 to 0.284 (**49.8%**) and MW from 63.917 to 9.799 (**84.7%**); STGG-50× achieves the best QED (**0.050**), while ours remains competitive (0.103). Diversity and uniqueness are high (Div ≈ 0.884, Uniq = 1.0), on par with the best graph baseline (GA-1000, Div = 0.886).

Table 2: Ablation study on retrieval and fragment-level optimizer (lower is better).

| Method | QED err | logP err | MW err | Norm. total err |
|---|---|---|---|---|
| Stage1 (no retrieval) | 0.111 | 0.970 | 68.555 | 0.307 |
| Stage1 + retrieval | **0.098** | 0.769 | 63.240 | 0.265 |
| Stage1 + retrieval + 1-hop | 0.130 | 0.423 | 10.404 | 0.187 |
| Stage1 + retrieval + 2-hop | 0.111 | 0.339 | 10.489 | 0.160 |
| Stage1 + retrieval + 3-hop | 0.103 | **0.284** | **9.799** | **0.146** |

## 4.3 ABLATION STUDY

**Interpretation.** Most LLMs show reasonable QED but large logP/MW errors (e.g., GPT-4.1 logP 0.697, MW 39.583), highlighting limited numeric control and multi-objection optimization. DrugAssist-7B even shows great repetition with uniqueness only 0.38. Graph search exhibits the opposite trade-off: STGG excels on QED but struggles on logP/MW; GA improves MW and diversity but retains higher logP (e.g., 0.529 for GA-1000). Our multi-hop refinement strikes the right

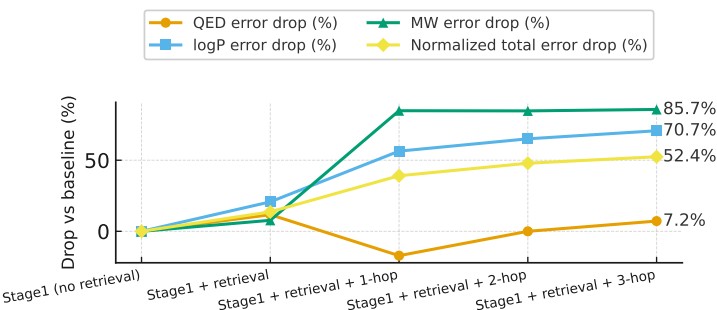

Figure 2: Ablation curves showing the *drop percentage* (higher is better) of each error metric relative to the no-retrieval baseline across methods. Curves are shown for QED, logP, MW, and the normalized total error.

balance, with NTE dropping from $0.187$ (1-hop) to $0.160$ (2-hop) to $0.146$ (3-hop), demonstrating controlled fragment-level edits that steadily minimize distance-to-target across properties.

We ablate three design choices on a held-out set: (i) Stage I without retrieval (baseline), (ii) Stage 1 with retrieval, and (iii) Stage I with retrieval followed by a fragment-level optimizer using 1/2/3 hops. We report per-property errors (QED, logP, MW) and the normalized total error ($e_{\mathrm{norm}} = |\Delta\mathrm{QED}| + |\Delta\log P|/10 + |\Delta\mathrm{MW}|/700$) in Table 2. For visualization, we plot the *drop percentage* relative to the no-retrieval baseline,

$$\mathrm{drop}(m) = \frac{e_{\mathrm{base}} - e_m}{e_{\mathrm{base}}} \times 100\%,$$

for each metric and method (Figure 2).

**Effect of retrieval** Adding retrieval already yields consistent gains: the normalized total error drops by **13.7%** ($0.307 \rightarrow 0.265$), driven primarily by improvements in logP (**20.7%** drop) and MW (7.8% drop). Retrieval also gives the best stand-alone QED error among non-optimized variants (**0.098**, **11.7%** drop).

**Effect of the fragment-level optimizer** Introducing the optimizer produces the largest improvements, especially on MW. Moving from retrieval-only to 1/2/3 hops reduces MW error from $\sim 63$ to $\sim 10$ (**84.9%** drop vs. baseline), and steadily improves logP (drops of $56.4\%$, $65.1\%$, and **70.7%**). The overall normalized error decreases monotonically with more hops: $0.187$ (1-hop, $39.1\%$ drop), $0.160$ (2-hop, $47.9\%$ drop), and **0.146** (3-hop, **52.4%** drop). QED exhibits a small regression at 1-hop (as expected when trading off multi-objective targets), but recovers by 3-hop to a $7.0\%$ drop versus baseline.

**Takeaway** Retrieval is a strong enabler, and the fragment-level optimizer is essential for precise multi-property alignment, culminating in the best overall performance with the 3-hop setting.

## 5 CONCLUSION AND LIMITATION

We introduced **M⁴olGen**, a two-stage, fragment-level framework for *precise, property-constrained* molecular generation. Stage I performs retrieval-augmented prototype construction; Stage II applies a GRPO-trained, multi-hop optimizer that explicitly minimizes distance-to-target while controlling edit complexity. A large, reasoning-ready dataset (BRICS fragments with neighbor pairs and measured property deltas) underpins both stages. Across QED, $\log P$, and MW targets, M⁴olGen attains the lowest normalized total error among strong LLM and graph baselines, with monotonic gains as hop count increases, and maintains high validity, uniqueness, and diversity. Taken together, these results validate our design choices and demonstrate the method's potential to scale to richer objectives.

While promising, our study is limited by its reliance on computed properties (e.g., RDKit estimators) and by the narrow property set evaluated (QED, Log P, MW). Going forward, we will broaden the objective space, support interval and Pareto objectives with uncertainty-aware rewards. We will also explore different reference models rather than RDkit.

## ETHICS STATEMENT

This work adheres to the ICLR Code of Ethics. In this study, no human subjects or animal experimentation was involved. All datasets used were sourced in compliance with relevant usage guidelines, ensuring no violation of privacy. We have taken care to avoid any biases or discriminatory outcomes in our research process. No personally identifiable information was used, and no experiments were conducted that could raise privacy or security concerns. We are committed to maintaining transparency and integrity throughout the research process.

## REPRODUCIBILITY STATEMENT

All codes have been made publicly available in an anonymous repository: `https://anonymous.4open.science/r/M4olgen-6FE2` to facilitate replication and verification. The experimental setup, including training steps, model configurations, and hardware details, is described in detail in the paper. The datasets and checkpoints will be released later due to size limitation.

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

# A APPENDIX

## A.1 STAGE I ALGORITHM

---

**Algorithm 1** Stage I: Local Optimal Candidate Generation via Multi-Agent Planning

---

**Require:** User query $q$, molecule database $\mathcal{M}$, thresholds $\epsilon_i$, max iterations $T$
1: $\mathbf{P}^* \leftarrow$ Decomposer($q$)
2: $\mathcal{N} \leftarrow$ Retriever($\mathbf{P}^*, \mathcal{M}, \epsilon_i$)
3: $m_0 \leftarrow$ InitialGeneration($q, \mathcal{N}, \mathbf{P}^*$)
4: Initialize reasoning history $\mathcal{H} \leftarrow []$
5: **for** $t = 1$ to $T$ **do**
6:      $a_t \leftarrow$ Reasoner($q, \mathcal{N}, \mathcal{H}, \mathbf{P}^*$)
7:      $m_t \leftarrow$ Edit($m_{t-1}, a_t$)
8:      $\mathcal{H} \leftarrow \mathcal{H} \cup \{a_t\}$
9:      **if** is_valid($m_t, \mathbf{P}^*, \epsilon_i$) **then**
10:          **return** $m_t$
11:      **end if**
12: **end for**
13: **return** valid $m_T$ if any

---

## A.2 END-TO-END DEMO: FROM LOCAL REASONER TO GRPO REFINEMENT

**Target.** We aim for **QED** $\approx 0.70$, **LogP** $\approx 1.50$, and **MW** $\approx 300$.

**Stage 1 — Iterative construction (LLM planner).** The planner begins from scratch and proposes *fragment-level* edits while reading back numeric feedback at each step.

*Step 1.* It proposes CCN(CC)C(=O)C(C1CC1)S(=O)=O based on relevant molecules, reasoning that a compact sulfonamide with small rings could balance QED and LogP. Feedback shows QED = 0.674 (below by 0.026), LogP = 0.245 (below by 1.255), MW = 219.306 (below by 80.694). The model decides to raise both LogP and MW.

*Step 2.* To add hydrophobic mass, it benzylates the amide nitrogen, yielding CCN(Cc1ccccc1)C(=O)C(C1CC1)S(=O)=O. Feedback: QED = 0.803 (above by 0.103), LogP = 1.425 (just 0.075 low), MW = 281.377 (still 18.623 low). The ring helped; MW needs a modest push upward.

*Step 3.* It enlarges the small ring to a cyclohexyl to push MW/LogP: CCN(Cc1ccccc1)C(=O)C(C1CCCCC1)S(=O)=O. Feedback: QED = 0.819 (high by 0.119), LogP = 2.595 (high by 1.095), MW = 323.458 (high by 23.458). Overshot both LogP and MW.

*Step 4.* It trims to cyclopentyl: `CCN(Cc1ccccc1)C(=O)C(C1CCCC1)S(=O)=O`. Feedback: QED = 0.820 (high by 0.120), LogP = 2.205 (high by 0.705), MW = 309.431 (high by 9.431). Still too heavy and too lipophilic.

*Step 5.* To temper LogP/MW while retaining aromaticity, it swaps phenyl → pyridine: `CCN(Cc1ncccc1)C(=O)C(C1CCCC1)S(=O)=O`. Feedback: QED = 0.811 (high by 0.111), LogP = 1.600 (high by 0.100), MW = 310.419 (high by 10.419). Closer on LogP, MW still a bit high.

*Step 6 (seed for Stage 2).* It reduces the ring to a butyl chain to lower MW/LogP: `CCN(Cc1ncccc1)C(=O)C(CCC)S(=O)=O`. Feedback: QED = 0.764 (high by 0.064), LogP = 1.210 (low by 0.290), MW = 284.381 (low by 15.619). This is the best Stage-1 candidate (normalized total error = 0.116) and becomes the seed for Stage 2.

**Stage 2 — GRPO refinement (accepted path with reasoning).** We now switch to the optimizer trained with GRPO. At each hop, we ask for a single fragment edit that moves QED/LogP/MW by specified deltas in the right directions, then accept only moves that improve the objective.

*Hop 1.* From the seed `CCN(Cc1ncccc1)C(=O)C(CCC)S(=O)=O`, we request: decrease QED by 0.064, increase LogP by 0.290, and increase MW by 15.619. **Reasoning.** The model replaces the sulfone side chain with a bicyclic, more drug-like fragment to add hydrophobic mass while modulating polarity. **Edit.** Replace `C(=O)C(CCC)[SH](=O)=O` → `C1=CNC(N)C(O)C=C(C)CC1=C`, producing `CCN(Cc1ncccc1)C1=CNC(N)C(O)C=C(C)CC1=C`. The move improves the objective and is *accepted*.

*Hop 2.* From that intermediate, we request: further decrease QED by 0.040, decrease LogP by 0.386, and decrease MW by 14.433. **Reasoning.** The optimizer softens hydrophobicity and trims mass while preserving the newly introduced scaffold connectivity. **Edit.** Replace `N()C1=CNC(N)C(O)C=C(C)CC1=C` → `NC1=CNNC=CC(O)CC(C)C1`, yielding `CCNC1=CNNC=CC(O)CC(C)C1Cc1ncccc1`. This further reduces the objective and is *accepted*.

**Final outcome.** The best molecule along this path is `CCNC1=CNNC=CC(O)CC(C)C1Cc1ncccc1` with QED = 0.681, LogP = 1.700, MW = 302.422, and a normalized total error of 0.042. In summary, Stage 1 quickly assembled a plausible prototype with sensible fragment choices, and Stage 2 applied two targeted, GRPO-guided edits that traded off hydrophobic mass and polarity to tighten alignment with all three numeric targets.

## A.3 USE OF LLMs

Large Language Models (LLMs) were used solely for writing refinement such as grammar and syntax improvements.

