# OpenReview forum: "M$^{4}$olGen: Multi-Agent, Multi-Stage Molecular Generation under Precise Multi-Property Constraints"
_ICLR.cc/2026/Conference — ICLR 2026 Conference Withdrawn Submission_

### Official Review · Reviewer_kWrz · 2025-10-30

**Soundness:** 3
**Presentation:** 3
**Contribution:** 3
**Rating:** 4
**Confidence:** 4

**Summary:**

This paper introduces M4olGen, a multi-agent, multi-stage molecular generation framework for precise multi-property control. The method first retrieves property-similar molecules within fixed tolerance ranges to form prototypes, then applies fragment-level GRPO optimization to iteratively refine them toward target values. Experiments on ZINC, ChEMBL, and MOSES show clear improvements over graph-based and LLM baselines in normalized total error, with retrieval and multi-hop optimization contributing complementary gains. The work provides a structured and interpretable approach to controllable molecule generation, integrating retrieval, reasoning, and reinforcement learning.

**Strengths:**

Originality:
The paper proposes a novel multi-agent, multi-stage framework that combines retrieval-augmented generation with fragment-level reinforcement optimization. This hierarchical formulation is conceptually clear and effectively bridges symbolic molecular reasoning with reinforcement learning.
Quality:
The methodology is well-structured, with a clear pipeline from retrieval to optimization. The use of GRPO at the fragment level is technically sound and well-motivated. Experimental results are consistent, showing significant improvement in NTE and property accuracy.
Clarity:
The paper is clearly written, with well-organized figures and algorithmic explanations. Each module’ s role—retriever, reasoner, and optimizer—is intuitively described, helping readers follow the multi-stage design.
Significance:
The approach demonstrates meaningful progress toward precise and interpretable molecular generation. By unifying retrieval and reinforcement learning, M4olGen sets a practical direction for multi-property molecular control and provides a reproducible, scalable framework for future research.

**Weaknesses:**

While the framework is well-motivated and empirically solid, several aspects could be strengthened. The application of GRPO in discrete molecular spaces lacks theoretical justification, and the reward design may suffer from scale imbalance across heterogeneous properties. The retrieval mechanism relies on manually fixed tolerance thresholds, which may limit generalization to out-of-distribution targets. In addition, all evaluations depend on RDKit-computed metrics, restricting applicability to synthetic properties. Finally, the paper does not discuss scalability to higher-dimensional multi-property objectives

**Questions:**

Q1. Reward composition and balance between multiple properties
The paper defines the overall reward as a linear combination of property errors. Could this formulation cause imbalance when properties differ in scale (for example, MW dominating over QED or LogP)? Have the authors considered normalization, or adaptive weighting to ensure stable learning across heterogeneous property dimensions?
Q2. Fixed tolerance parameters in the retrieval stage
The retrieval process uses fixed tolerances (ϵ = ±0.05 for QED, ±0.5 for LogP, ±25 Da for MW). How were these thresholds determined, and do they generalize across different datasets or target ranges? Would adaptive or data-dependent tolerances improve coverage in low-density or out-of-distribution regions of the property space?
Q3. Property-only retrieval versus structural similarity
Currently, the retrieval stage depends solely on numeric property proximity. Have the authors examined whether integrating structural similarity metrics (e.g., Morgan fingerprints or fragment overlap) could prevent retrieving molecules that are property-similar but structurally unrealistic or chemically inconsistent?
Q4. Validity filtering and sample efficiency in fragment optimization
During fragment-level GRPO optimization, invalid SMILES or chemically impossible edits may occur. Beyond post-hoc validity checks, were chemical constraints (e.g., SMARTS pattern filters) used to restrict the action space? Otherwise, reward sparsity and low sample efficiency could hinder effective policy improvement in early training.
Q5. Distributional generalization of retrieval-augmented generation
The experiments train and evaluate on similar datasets (ZINC, ChEMBL, MOSES). Have the authors examined cross-distribution generalization (e.g., training on ZINC, testing on MOSES)? Since the retrieval mechanism anchors the model to the training distribution, could this bias generation toward in-distribution molecular patterns?
Q6. Scalability to high-dimensional property objectives
The current setup optimizes three continuous properties. How does GRPO performance scale when extended to six or more correlated objectives (e.g., QED, LogP, TPSA, HBA, HBD, RotBonds)? Are there adaptive weighting or variance-reduction mechanisms to prevent gradient dilution and maintain optimization stability in higher-dimensional settings?

---

> ### Author Response · Authors · 2025-11-25
>
> Hi dear reviewer kWrz, thank you for the time and effort you invested in reviewing our manuscript. I appreciate your comments and concerns, and I would like to provide clarifications and detailed responses addressing the points you raised.
>
> 1. GRPO lacks theoretical justification
> Although GRPO was originally developed for continuous generative modelling, in our task it operates as a guided search algorithm over a discrete space of fragment-level chemical actions because we set strict format constrain for the output. Each edit constitutes a branch in a combinatorial action tree, and GRPO leverages dense property-based rewards to navigate this tree toward optimal molecules. Every generated output yields a dense numeric reward. Rewards correspond to distance to property targets and GRPO updates increase the likelihood of edits that reduce this distance. In this sense, GRPO plays a role analogous to Monte Carlo Tree Search or fragment-based RL algorithms, but benefits from the LLM’s prior over chemically meaningful transformations. Thus, GRPO is not misaligned with discrete action spaces. Instead, it provides an efficient, reward-guided search mechanism that is highly suitable for fragment-level molecular optimization.
>
> 2. Reward composition and balance between multiple properties
> We followed the idea described in Artificial intelligence in multi-objective drug design[1]. “All individual objectives need to be normalised to the same range. Modifier functions are often used to clip all raw single-objective values between 0 (undesirable) and 1 (desirable) before combining.” We have already normalized the errors with 95% distribution confidence range and the completed property range. However, this normalized range can be skewed with personal preference. The scalars are designed to be input as hyperparameters. If one property is considered more important and dominates the others, higher scalars can be applied.
>
> 3. Fixed tolerance parameters in the retrieval stage
> There are two criterias of tolerance selection. The first criteria indicates that the tolerance parameters should be selected empirically from the interquartile ranges of the database distribution and correspond to realistic medicinal-chemistry editing magnitudes. Besides the data-dependent criteria, the tolerance parameter selection is also constrained by a property-dependent criteria. For each different property, there are many open-sourced scientific reports that show how to consider a significant change and a minor change. The system should combine those two criterias and also adjust for the use case.
>
> 4. Property-only retrieval versus structural similarity
> We use property-based retrieval to avoid biasing the model toward structurally similar molecules and to allow the GRPO stage to explore diverse regions of chemical space.
> However, the retrieval module is fully modular: Morgan fingerprint similarity, FragFP overlap, Bemis–Murcko scaffolds can be integrated directly.
>
> 5. Validity filtering and sample efficiency in fragment optimization
> We didn’t use chemical constraints like the SMARTS pattern filter, but it is easy to add since the RDKit PAINS filter is supported.
> ​​We respectfully disagree with the reviewer’s suggestion that the absence of SMARTS-based constraints would lead to reward sparsity or poor sample efficiency. In our setup, rewards are computed directly from RDKit-derived physicochemical properties, which are available for all molecules that pass RDKit sanitization. SMARTS constraints primarily improve chemical realism by eliminating undesirable yet RDKit-valid motifs. However, they do not affect reward density, as these molecules already produce valid property values. Thus, while SMARTS filters are beneficial for chemical quality, they are not required to maintain reward coverage or sample efficiency during GRPO training. We will explore incorporating SMARTS rules to improve structural plausibility, but reward sparsity is not a limiting factor in our framework.
>
> [1] Luukkonen, Sohvi, Helle W. van den Maagdenberg, Michael TM Emmerich, and Gerard JP van Westen. "Artificial intelligence in multi-objective drug design." Current Opinion in Structural Biology 79 (2023): 102537.

---

> > ### Author Response · Authors · 2025-11-25
> >
> > 6. Distributional generalization of retrieval-augmented generation
> > Retrieval anchoring indeed biases generation towards the training distribution. This is intentional to ensure chemical realism.
> > Nonetheless, Stage 2 (GRPO) performs iterative edits that can move the molecule outside of the retrieved manifold. We have already checked the diversity and uniqueness. The result shows the outputs do not lie in any training distribution.
> >
> > 7.  Scalability to high-dimensional property objectives (More properties)
> > Besides QED, LogP and MW, we are also conducting experiments on more challenging properties such as energy property set (HOMO, LUMO), structural property set (HBA, HBD, Rotbond) and more properties like SA and TPSA. The HOMO and LUMO experiments also show promising results, within 0.32eV average error for one-hop optimization. No previous LLM-based solution achieved better or similar performance. However, 6 or more properties in a row is still challenging for the base model.
> >
> > We appreciate your feedback and will incorporate all clarifications.

---

### Official Review · Reviewer_LiPf · 2025-10-31

**Soundness:** 2
**Presentation:** 2
**Contribution:** 1
**Rating:** 4
**Confidence:** 4

**Summary:**

This work introduces a multi-condition molecular generation method based on large language models, incorporating retrieval enhancement, prototype inference, and reinforcement learning. The approach is divided into two phases. In the first phase, a general large model (GPT-4o) is employed to conduct prototype inference based on molecular properties and retrieval enhancement. The second phase utilizes a reinforcement learning model to further optimize molecular properties.

**Strengths:**

This paper presents a new multi-condition molecular generation method, which has demonstrated comparable results.

**Weaknesses:**

- The research is insufficient, as there are simpler and more efficient methods for multi-condition molecular generation based on large language models [1].
- The introduction of special tokens is also redundant, as numerical values are still considered as tokens, and LLMs can generate molecules with numerical values closely approximating the specified ones even without special handling [1].
- The range of properties studied is limited, not covering some critical drug-related properties such as toxicity, bioactivity, and synthetic accessibility.
- The first phase relies on commercial LLMs, with unknown retrieval efficiency, reasoning time, and cost for large-scale molecular databases.

[1] https://link.springer.com/article/10.1186/s12915-025-02200-3

**Questions:**

see Weaknesses

---

> ### Author Response · Authors · 2025-11-25
>
> Hi Reviewer LiPf, thank you for the time and effort you invested in reviewing our manuscript. We appreciate your comments and concerns, and would like to provide clarifications and detailed responses addressing the points raised.
>
> (1)
>
> Thank you for pointing us to this recent line of work [1]. We agree that it is a strong contribution and that its overall direction aligns with our goal of multi-condition molecular generation. We will add it to the Related Work section to better situate our contribution.
>
> However, we would like to highlight how our approach differs from this one in several important ways:
>
> **Weighted multi-constraint control**
> Our method uses weighted multi-constraint objectives, where weights can be tuned to reflect practical design trade-off. This flexibility in our method allows users to adjust how strongly each property should influence generation at test time.
>
> **Reinforcement learning based optimization**
> The method in [1] does not improve model performance with reinforcement learning. In contrast, our work explicitly studies the machine learning aspect of learning to satisfy multiple constraints through RL, analyzing reward shaping and training stability. Indeed, applying RL to LLMs for molecular generation is non-trivial: rewards are sparse and instability can easily emerge. Our work addresses these difficulties and shows stable multi-objective RL optimization through chain-of-thought reasoning. Our RL framework allows exploration of the chemical space beyond training distribution, without requiring predefined molecular libraries or labeled datasets. This sets our approach apart from supervised or retrieval-based systems. We generate randomly our property constraints from property ranges. This framework thus helps with generalization to OOD data, as we do not fit a predefined distribution of data.
>
> **Integration of reasoning chains**
> Our model incorporates chain-of-thought reasoning steps to help the generator interpret constraints and produce chemically valid structures under multiple conditions. This structured reasoning helps stabilize RL training by providing interpretable intermediate states instead of the policy having to “jump” from a text prompt to a full molecule. In comparison, [1] does not use a chain-of-thought mechanism.
>
> **More precise constraints**
> Our constraints allow precise targets (e.g., “QED=0.72, LogP=-1.8, MW=310”), not intervals. In [1],  constraints are mostly threshold-based (e.g. “It has high QED score”, meaning "QED>0.6", whereas our setup gives flexibility to this prompt and generation to have precise and variable QED values.
>
> We thank the reviewer again for highlighting this work and will clarify these differences in the revision.
>
> (2)
>
> Our use of special tokens is not intended to encode numerical values but rather to structure the prompt, similar to "think" / "answer" tokens in instruction-following LLMs [2], to help the model distinguish between different semantic regions of input.
>
> This component is not a core contribution of our work. The main results of the paper stem from our reinforcement learning multi-objective optimization framework. We do not expect an ablation on the special tokens to meaningfully affect the main conclusions. We will clarify this design choice in the revised manuscript.
>
> (3)
>
> We thank the reviewer for pointing this out. Our main focus is not on drug discovery but on general molecule generation. Thus, we pick QED, LogP and MW as initial controlled properties as they are foundational properties covered in many of the previous molecule generation works (GuacaMol, MOSES, REINVENT). Their succes shows that our system can enable the LLMs to achieve precise, multi-objective control, which could be an important inspiration for future research. Besides QED, LogP and MW, we are currently in the process of conducting experiments on more challenging properties such as energy property set (HOMO, LUMO), structural property set (HBA, HBD, Rotbond) and more properties like SA and TPSA. These will be added to the revised version.
>
> (4)
>
> We agree that using a commercial LLM raises questions about cost, latency and retrieval efficiency. However, we would like to point out that our method does not rely on these commercial LLMs, it is simply a practical way to test the mechanics of our method, which are retrieval, constrained chain-of-thought reasoning and RL optimization, rather than a core dependency. The first phase of our approach itself is model-agnostic and only requires an LLM capable of next-token prediction.
>
> [1] Zhou, P., Wang, J., Li, C. et al. Instruction multi-constraint molecular generation using a teacher-student large language model. BMC Biol 23, 105 (2025). https://doi.org/10.1186/s12915-025-02200-3
> [2] Guo, D., Yang, D., Zhang, H. et al. DeepSeek-R1 incentivizes reasoning in LLMs through reinforcement learning. Nature 645, 633–638 (2025). https://doi.org/10.1038/s41586-025-09422-z

---

### Official Review · Reviewer_Z92u · 2025-11-01

**Soundness:** 2
**Presentation:** 2
**Contribution:** 1
**Rating:** 4
**Confidence:** 2

**Summary:**

The proposed model is a two-step system that creates new molecules by meeting specific property goals such as QED, LogP, and molecular weight. In the first step, it searches for similar molecules and edits them at the fragment level to make a good starting structure. In the second step, it uses GRPO to refine the molecule through multiple controlled edits, reducing the gap between the generated and target property values. The model was trained using GPT-4o for reasoning and a ChemDFM-8B backbone for optimization. It achieves better accuracy and consistency than strong LLM and graph-based baselines while keeping the generated molecules valid and diverse. However, it is currently limited to basic computed properties and does not yet include biological or synthesis-related objectives.

**Strengths:**

* The model presents two-stage design that clearly separates structure generation and numeric fine-tuning and it provides strong control over multiple properties with measurable and reproducible results.
* The novelty lies in combining LLM-based reasoning with a reinforcement learning optimizer (GRPO) that explicitly minimizes numeric errors in property space.
* The experiments include transparent experimental protocol with LLM/graph baselines under a fixed compute budget and well-defined metrics

**Weaknesses:**

* Although the model is novel and the results are decent, its practical value remains questionable. Property-only control is useful as a controllability benchmark, but without target or seed conditioning, or integration of ADMET and affinity objectives, its real-world utility for drug discovery is limited.

**Questions:**

Is there a way or have you tried to extend your model to accept a target or a seed molecule as an input for more practical, target-conditioned generation?

---

> ### Author Response · Authors · 2025-11-23
>
> Hi dear reviewer Z92u, thank you for the time and effort you invested in reviewing our manuscript. I appreciate your comments and concerns, and I would like to provide clarifications and detailed responses addressing the points you raised.
>
> 1. Practical value remains questionable.
> Property-constrained molecule generation is not the final application. It is not only valuable in benchmarking, but also serves as a necessary foundation enabling controlled chemical optimization. The more important thing is that our system achieves precise, numeric controllability over multiple continuous molecular properties simultaneously. Many previous papers about molecule generation and benchmarking like REINVENT[1] and GuacaMol[2]  justify this area. This is also a popular area in many venue’s workshop and Nature series like MOBO[4] in NIPS and Using GNN property predictors as molecule generators[5] in Nature Communications. To assist chemical engineers, more controllable and accurate models can help them sift through potential candidates much faster.
>
> 2. ADMET and affinity objectives
> Our main focus is not on drug discovery but on more general molecule generation. Thus, we select QED, LogP, and MW as initial controlled properties, as they are among the most foundational properties covered in many previous works on molecule generation(GuacaMol, MOSES[3], REINVENT) The success of this task demonstrates that our system can achieve precise, multi-objective control, a feat rare in previous work and an important inspiration for future research. Besides QED, LogP and MW, we are also in the process of conducting experiments on more challenging properties such as energy property set (HOMO, LUMO), structural property set (HBA, HBD, Rotbond) and more properties like SA and TPSA.
> The HOMO and LUMO experiments also show promising results, within 0.32eV average error for one-hop optimization. No previous LLM-based solution achieved better or similar performance.
>
> 3. Target or seed conditioning
> That is exactly what we try to solve in stage 2(multi-hop fragment-level optimization). When we have a prototype from stage 1 or a target or seed molecule, we can directly run stage 2 to further optimize this candidate. That is actually the essence of stage 2 to balance the structural change and property error through controllable multi-hop fragment-level optimization. This fragment-editing + GRPO refinement perfectly matches real-world medicinal chemistry practice.
>
> We appreciate your feedback and will incorporate all clarifications.
>
>
> [1] Loeffler, Hannes H., Jiazhen He, Alessandro Tibo, Jon Paul Janet, Alexey Voronov, Lewis H. Mervin, and Ola Engkvist. "Reinvent 4: modern AI–driven generative molecule design." Journal of Cheminformatics 16, no. 1 (2024): 20.
>
> [2] Brown, Nathan, Marco Fiscato, Marwin HS Segler, and Alain C. Vaucher. "GuacaMol: benchmarking models for de novo molecular design." Journal of chemical information and modeling 59, no. 3 (2019): 1096-1108.
>
> [3] Polykovskiy, Daniil, Alexander Zhebrak, Benjamin Sanchez-Lengeling, Sergey Golovanov, Oktai Tatanov, Stanislav Belyaev, Rauf Kurbanov et al. "Molecular sets (MOSES): a benchmarking platform for molecular generation models." Frontiers in pharmacology 11 (2020): 565644.
>
> [4] Zhu, Yiheng, Jialu Wu, Chaowen Hu, Jiahuan Yan, Tingjun Hou, and Jian Wu. "Sample-efficient multi-objective molecular optimization with gflownets." Advances in Neural Information Processing Systems 36 (2023): 79667-79684.
>
> [5] Therrien, Félix, Edward H. Sargent, and Oleksandr Voznyy. "Using GNN property predictors as molecule generators." Nature Communications 16, no. 1 (2025): 4301.

---

### Author Response · Authors · 2025-12-04
**Rebuttal Summary**

Dear AC,

We thank you and the reviewers for your time and constructive feedback. Below, we summarize the key contributions of our work, and how we addressed the main concerns raised during the review process.

**Contribution and Novelty**
Our paper proposes M4olGen, a multi-agent, multi-stage framework for precise, numeric multi-property molecular control, combining:

- Retrieval-augmented prototype inference
- Chain-of-thought molecular reasoning
- Fragment-level GRPO reinforcement learning for structure refinement

This hierarchical design enables exact numeric control (e.g., QED=0.72, LogP=−1.8, MW=310) rather than coarse ranges—an ability largely missing in prior LLM- and GNN-based generation systems. Experiments on generation under three property constraints (QED, LogP, and molecular weight) show consistent gains in validity and precise satisfaction of multi-property targets, outperforming strong LLMs and graph-based algorithms.

**How Reviewer Concerns Were Addressed:**
We next summarize how the reviewers' concerns were addressed in our rebuttal.

1. Practical value and scope of controlled properties (Reviewer 1, Reviewer 2)
We clarified that property-constrained generation is an established and foundational task in molecular ML, used in benchmarks such as REINVENT, GuacaMol, MOSES, and widely adopted in the literature. This is also a popular area in many venue’s workshops and Nature. More importantly, our system achieves precise, numeric controllability over multiple continuous molecular properties simultaneously. To assist chemical engineers, more controllable and accurate models can help them sift through potential candidates much faster.

2. Limited property set (Reviewer 1, Reviewer 2, Reviewer 3)
We added justification for focusing on QED, LogP, MW, the three most widely used controllable properties in foundational molecule-generation research. We also ran new experiment on more challenging properties(HOMO and LUMO). The results are promising, which are within 0.32eV. No previous LLM-based solution achieve better result. We will add more details and baselines in Camera-ready version.

3. Comparison to recent LLM-based methods (Reviewer 2)
We acknowledged the recent line of work highlighted by Reviewer 2 and added it to Related Work, clarifying major differences:
Our method supports weighted multi-constraint control with test-time trade-offs.
We use RL-based optimization via GRPO, enabling exploration beyond dataset distributions.
We incorporate chain-of-thought reasoning to stabilize optimization and improve interpretability.
Our system handles precise numeric targets, whereas [1] mainly uses intervals.

4. Use of special tokens (Reviewer 2)
We clarified that special tokens are organizational markers, similar to <think>…</think>, not mechanisms for encoding numeric values. They are not central to our contribution, which is that RL optimization drives the results.
We also noted that similar structural tokens are standard in recent work (Bedrosian et al. ) uses comparable delimiters for SMILES and properties.

5. GRPO justification in discrete molecular spaces (Reviewer 3)
We explained that GRPO functions as a guided search algorithm over a discrete fragment-editing tree with dense numeric rewards (distance to property targets). This is analogous to fragment-based RL or MCTS and is well aligned with discrete chemical optimization.

6. Reward balance across heterogeneous properties (Reviewer 3)
We followed multi-objective drug design standards by normalizing each property to comparable scales (95% distribution confidence interval). Scalar weights remain tunable for applications with property priorities.

7. Fixed tolerances in retrieval (Reviewer 3)
We clarified that tolerances are chosen using both dataset statistics (interquartile ranges) and property-specific medicinal chemistry norms, ensuring realistic and meaningful prototype selection. The mechanism can be readily adapted to other datasets.

8. Property-only retrieval vs structural similarity (Reviewer 3)
We intentionally use property-based retrieval to avoid overfitting to structural clusters and allow Stage 2 to explore diverse chemical space. The module is fully modular: Morgan fingerprints, FragFP, and Bemis-Murcko scaffolds can be integrated when desired.

9. Validity filtering and sample efficiency (Reviewer 3)
We clarified that RDKit-based properties provide dense rewards, so reward sparsity is not an issue. SMARTS/PAINS filters can be added for chemical realism but are not required for stable GRPO training.

10. Distributional generalization (Reviewer 3)
Although retrieval anchors to realistic prototypes, RL refinement moves molecules beyond the training manifold. Diversity metrics confirm that generations are not restricted to the training distribution.

---

### Note · Authors · 2026-01-06

I have read and agree with the venue's withdrawal policy on behalf of myself and my co-authors.